# Ovarian Adnexal Reporting Data System (O-RADS) for Classifying Adnexal Masses: A Systematic Review and Meta-Analysis

**DOI:** 10.3390/cancers14133151

**Published:** 2022-06-27

**Authors:** Julio Vara, Nabil Manzour, Enrique Chacón, Ana López-Picazo, Marta Linares, Maria Ángela Pascual, Stefano Guerriero, Juan Luis Alcázar

**Affiliations:** 1Department of Obstetrics and Gynecology, Clínica Universidad de Navarra, 31008 Pamplona, Spain; jvara@unav.es (J.V.); nmanzour@unav.es (N.M.); echaconc@unav.es (E.C.); alopezpicazom@unav.es (A.L.-P.); 2Department of Obstetrics and Gynecology, Universitiy Hospital Puerta del Mar, 11009 Cadiz, Spain; martalinaresregidor@gmail.com; 3Department of Obstetrics, Gynecology, and Reproduction, Hospital Universitari Dexeus, 08028 Barcelona, Spain; marpas@dexeus.com; 4Department of Obstetrics and Gynecology, University of Cagliari, Policlinico Universitario Duilio Casula, 09042 Monserrato, Cagliari, Italy; gineca.sguerriero@tiscali.it

**Keywords:** ovarian neoplasms, ovarian cancer diagnosis, malignancy, benign neoplasms, O-RADS, meta-analysis, transvaginal ultrasound

## Abstract

**Simple Summary:**

We performed a systematic review and meta-analysis aiming to assess the diagnostic performance of the Ovarian Adnexal Report Data System (O-RADS) using transvaginal ultrasound for classifying adnexal masses. Data from 11 studies comprising 4634 masses showed that the pooled estimated sensitivity, specificity, positive likelihood ratio, negative likelihood ratio, and diagnostic odds ratio of O-RADS system for classifying adnexal masses were 97% (95% confidence interval (CI) = 94%–98%), 77% (95% CI = 68%–84%), 4.2 (95% CI= 2.9–6.0), 0.04 (95% CI = 0.03–0.07), and 96 (95% CI = 50–185), respectively. We concluded that the O-RADS system has good sensitivity and moderate specificity for classifying adnexal masses.

**Abstract:**

In this systematic review and meta-analysis, we aimed to assess the pooled diagnostic performance of the so-called Ovarian Adnexal Report Data System (O-RADS) for classifying adnexal masses using transvaginal ultrasound, a classification system that was introduced in 2020. We performed a search for studies reporting the use of the O-RADS system for classifying adnexal masses from January 2020 to April 2022 in several databases (Medline (PubMed), Google Scholar, Scopus, Cochrane, and Web of Science). We selected prospective and retrospective cohort studies using the O-RADS system for classifying adnexal masses with histologic diagnosis or conservative management demonstrating spontaneous resolution or persistence in cases of benign appearing masses after follow-up scan as the reference standard. We excluded studies not related to the topic under review, studies not addressing O-RADS classification, studies addressing MRI O-RADS classification, letters to the editor, commentaries, narrative reviews, consensus documents, and studies where data were not available for constructing a 2 × 2 table. The pooled sensitivity, specificity, positive and negative likelihood ratios, and diagnostic odds ratio (DOR) were calculated. The quality of the studies was evaluated using QUADAS-2. A total of 502 citations were identified. Ultimately, 11 studies comprising 4634 masses were included. The mean prevalence of ovarian malignancy was 32%. The risk of bias was high in eight studies for the “patient selection” domain. The risk of bias was low for the “index test” and “reference test” domains for all studies. Overall, the pooled estimated sensitivity, specificity, positive likelihood ratio, negative likelihood ratio, and DOR of the O-RADS system for classifying adnexal masses were 97% (95% confidence interval (CI) = 94%–98%), 77% (95% CI = 68%–84%), 4.2 (95% CI = 2.9–6.0), 0.04 (95% CI = 0.03–0.07), and 96 (95% CI = 50–185), respectively. Heterogeneity was moderate for sensitivity and high for specificity. In conclusion, the O-RADS system has good sensitivity and moderate specificity for classifying adnexal masses.

## 1. Introduction

Accurate discrimination between benign and malignant adnexal masses is essential for adequate management. Adnexal lesions considered as physiologic processes or at low risk of malignancy can be managed expectantly or removed through minimally invasive techniques [1,2]. On the other hand, adnexal masses classified at high risk of malignancy warrant further evaluation and should be eventually referred to gynecologic oncology units for adequate management [3].

Transvaginal sonography (TVS) is still considered the first-line imaging technique for assessing adnexal masses, and no other imaging technique provides better diagnostic performance than TVS [4]. Traditionally, TVS assessment of adnexal masses is based on the subjective impression of an expert examiner using so-called pattern recognition [5]. However, this approach is mainly limited by the need for proper training and experience [6]. For this reason, the use of several approaches such as ultrasound-based scoring systems, mathematical models, or biomarker-based models has been proposed [7,8,9,10,11,12]. Recently, the Assessment of Different Neoplasias in the Adnexa (ADNEX) model was proven to be the best model for discriminating benign from malignant adnexal masses [13]. An additional problem resides in the reporting of data. There is evidence that this problem may influence patient management [14].

In 2020, the American College of Radiology developed the so-called Ovarian Adnexal Report Data System (O-RADS) for classifying adnexal masses, with the aim of decreasing or eliminating ambiguity related to ultrasound reports [15]. This reporting system classifies adnexal masses into five risk groups (O-RADS 1: risk of malignancy 0%; O-RADS 2, risk of malignancy <1%; O-RADS 3, risk of malignancy 1–9%; O-RADS-4, risk of malignancy 10–49%; O-RADS 5, risk of malignancy ≥50%) on the basis of both the International Ovarian Tumor Analysis group (IOTA) terms and definitions and the ADNEX model. Since this report, several studies have been published addressing the diagnostic performance of this classification system. The aim of the present systematic review and meta-analysis was to synthesize the current evidence of the O-RADS reporting system for classifying adnexal masses.

## 2. Materials and Methods

### 2.1. Protocol and Registration

This meta-analysis was performed following the recommendations of the PRISMA Statement (http://www.prisma-statement.org/, accessed on 24 April 2022), as well as the guidelines from the Synthesizing Evidence from Diagnostic Accuracy Tests (SEDATE) [16,17]. The protocol was not registered. Inclusion and exclusion criteria for studies to be selected, as well as how data extraction and quality assessment were defined, were established prior to starting the data search.

Institutional Review Board approval from Clinical Universidad de Navarra was waived because of the study’s nature and design.

### 2.2. Data Sources and Searches

Three of the authors screened five electronic databases, PubMed/Medline, Scopus, Cochrane, Web of Science, and Google Scholar, to identify potentially eligible studies published between January 2020 and April 2022.

The search terms included and captured the concepts of “adnexal masses”, “ovarian cancer”, “transvaginal ultrasound”, “O-RADS”, and/or “Ovarian Adnexal Report Data System”. No language limit was set.

### 2.3. Study Selection and Data Collection

Two authors screened the titles and abstracts identified by the search to exclude irrelevant articles. Then, full-text articles were selected to identify potentially eligible studies applying the following three inclusion criteria:(1)Prospective and retrospective cohort study including patients diagnosed as having at least one adnexal mass classified using the O-RADS system after transvaginal/transabdominal ultrasound assessment as the index test.(2)Report of histologic diagnosis of the adnexal mass after surgical removal or conservative management demonstrating spontaneous resolution or persistence in cases of benign appearing masses after follow-up scan as the reference standard.(3)Presence of data reported that would allow constructing a 2 × 2 table to estimate true positive, true negative, false positive, and false negative cases for the O-RADS system.

Exclusion criteria were as follows: studies not related to the topic under review, studies not addressing O-RADS classification, studies addressing MRI O-RADS classification, letters to the editor, commentaries, narrative reviews, consensus documents, and studies where no data were available for constructing a 2 × 2 table.

For avoiding the inclusion of duplicate cohorts from at least two studies reported from the same authors, the study period of each study was examined; if dates overlapped, we contacted the authors to check whether the same patients were included in the different studies reported. We searched for additional papers by reading the reference lists of those papers selected for full-text reading. In cases of insufficient data, we contacted the authors. The Patients, Intervention, Comparator, Outcomes, Study Design (PICOS) criteria used for inclusion and exclusion of studies were recorded.

Two authors independently retrieved the diagnostic accuracy results from the ultimately selected studies. Disagreements arising during the process of study selection and data extraction were resolved by consensus between these two authors.

### 2.4. Risk of Bias in Individual Studies

Quality assessment of studies included in the meta-analysis was conducted using the tool provided by the Quality Assessment of Diagnostic Accuracy Studies-2 (QUADAS-2) [18]. The QUADAS-2 format includes four domains: (1) patient selection, (2) index test, (3) reference standard, and (4) flow and timing. For each domain, the risk of bias and concerns about applicability (the latter not applying to the domain of flow and timing) were analyzed and rated as low, high, or unclear risk. Quality assessment was used to provide an evaluation of the overall quality of the studies and to investigate potential sources of heterogeneity.

Two authors independently evaluated the methodological quality. Disagreements were solved by discussion between these authors. The assessment of the quality was based on whether the study described the study’s design, as well as inclusion and exclusion criteria, for the patient selection domain (studies with inadequate exclusions, retrospective studies with examiners not blinded to reference standard, and studies mixing data from expert and nonexpert examiners were considered at high risk of bias), whether the study reported on how the of the index test was performed and interpreted for the index test domain (studies not reporting whether the O-RADS classification was established using IOTA descriptors of the mass or the ADNEX model were considered at high risk), which was the reference standard used for the reference standard domain (for this domain, in case of conservative management of the mass, at least 1 year of follow-up was considered as appropriate to identify true negative or false negative cases), and description of the time elapsed from index test assessment to the reference standard result for the flow and timing domain (surgery >180 days after diagnosis was considered as high-risk). Unclear risk was stated when the corresponding information for each domain was not reported in the study.

### 2.5. Statistical Analysis

We extracted information on the diagnostic performance of the O-RADS system. O-RADS classifies the adnexal masses into five groups (see above), O-RADS 1, O-RADS 2, O-RADS 3, O-RADS 4, and O-RADS 5, on the basis of either IOTA terms for description of the mass or the ADNEX model. We used the following dichotomous classification for constructing the 2 × 2 tables: O-RADS 1–3 cases were considered as benign, and O-RADS 4–5 cases were considered as malignant.

A random effects model was used to estimate the pooled sensitivity, specificity, positive likelihood ratio (LR+), negative likelihood ratio (LR−), and diagnostic odds ratio (DOR). Likelihood ratios were used to characterize the clinical utility of a test and to estimate the post-test probability of disease [19]. In cases where a study reported data from the same cohort analyzed by expert and nonexpert examiners, we chose the data derived from expert examiners.

Using the mean prevalence of ovarian malignancy (pre-test probability), post-test probabilities were calculated using the positive and negative likelihood ratios and plotted on Fagan’s nomogram.

Heterogeneity for sensitivity and specificity was assessed using Cochran’s Q statistic and the I^2^ index [20]. A *p*-value < 0.1 indicates heterogeneity. I^2^ values of 25%, 50%, and 75% would be considered to indicate low, moderate, and high heterogeneity, respectively [20]. Forest plots of sensitivity and specificity of all studies were plotted. Meta-regression was used if heterogeneity existed for assessing covariates that could explain this heterogeneity. The covariates analyzed were sample size and malignancy prevalence.

Summary receiver operating characteristic (sROC) curves were plotted to illustrate the relationship between sensitivity and specificity. Lastly, publication bias was assessed using Deek’s method [21].

All analyses were performed using MIDAS and METANDI commands in STATA version 12.0 for Windows (Stata Corporation, College Station, TX, USA). A *p*-value < 0.05 was considered statistically significant.

## 3. Results

### 3.1. Search Results

The electronic search provided 502 citations. After excluding 101 duplicate records, 401 citations remained. After reading titles and abstracts, 387 citations were excluded (papers not related to the topic (*n* = 276), reviews (*n* = 74), papers assessing O-RADS MRI (*n* = 27), and case reports, letters to the editor, and commentaries (*n* = 10)). Fourteen papers remained for full-text reading. After full-text reading, three papers were excluded (no data for constructing 2 × 2 table available (*n* = 2), and use of MRI, instead of ultrasound (*n* = 1)). Eleven papers were ultimately included in the qualitative and quantitative synthesis [22,23,24,25,26,27,28,29,30,31,32]. A flowchart summarizing the literature search is shown in Figure 1.

### 3.2. Characteristics of Included Studies

As stated above, 11 studies published between February 2021 and April 2022 reporting on 4634 adnexal masses in 4525 women were included in the final analyses. Out of these 4634 adnexal masses, 1404 were malignancies (including 268 borderline tumors, 1058 primary ovarian cancers, and 78 metastatic cancers). In one study, four cases were considered as malignant for the final outcome on the basis of tumor growth during surveillance, but no histology was reported (30). According to histology, primary cancers were distributed as epithelial carcinomas (*n* = 829), nonepithelial tumors (*n* = 94), and “other malignant primary malignant lesions (*n* = 131). A total of 2827 benign tumors were removed surgically; the more frequent histologic types were as follows: dermoid cysts (*n* = 907, 32.1%), endometriomas (*n* = 704, 24.9%), serous/mucinous cystadenomas (*n* = 635, 22.5%), hydrosalpinx/tubo-ovarian abscesses (*n* = 137, 4.8%), functional cysts (*n* = 95, 3.4%), fibromas/fibrothecomas/thecomas (*n* = 88, 3.1%), para-ovarian cysts (*n* = 73, 2.6%), hemorrhagic cysts (*n* = 47, 1.7%), cystadenofibromas (*n* = 38, 1.3%), and other benign tumors (*n* = 103, 3.6%). One paper reported on 25 cases of adnexal masses with two or more different histological diagnoses in the same adnexal mass (so-called “collision tumors”) [31].

The mean prevalence of ovarian malignancy was 32%, ranging from 8% to 70%. All studies reported the clinical characteristics of the cohort to some extent. All studies reported the mean age of patients and distribution according to menopausal status. In fact, seven studies reported the distribution of benign and malignant tumors according to menopausal status [23,24,25,27,28,29,32]. According to these studies, 402 out of 2379 (16.9%) premenopausal women had malignant lesions, whereas 474 out of 832 (56.8%) postmenopausal women had malignant lesions.

Table 1 shows the PICOS features of the studies included.

All studies reported the distribution of masses according to O-RADS classification and the reference standard (Table 2). The prevalence of malignant cases for O-RADS 1–2, O-RADS 3, O-RADS 4, and O-RADS 5 was 1.0%, 4.5%, 45.7%, and 87.3%, respectively.

Four studies reported the O-RADS distribution according to the histology of surgically removed tumors, accounting for a total of 1964 lesions [23,24,25,31]. Table 3 shows this distribution.

### 3.3. Methodological Quality of Included Studies

The QUADAS-2 assessment of the risk of bias and concerns regarding applicability of the selected studies is shown graphically in Figure 2.

The study design was retrospective in 10 studies [22,23,24,25,26,27,29,30,31,32] and prospective in just one study [28]. Eight studies were considered as having high risk regarding the patient selection domain, since inappropriate exclusions (for example, cases with poor image quality or cases with not all data available) were observed [22,23,25,27,28,29,30,31,32], and one study was unclear since a complete description of the exclusion criteria was lacking [26].

Concerning the domain “index test”, all studies adequately described the method of the index text, as well as how it was performed and interpreted. Ten studies used IOTA terminology of the mass features [22,23,25,26,27,28,29,30,31,32], while one used the ADNEX model [24]. In all retrospective studies, examiners were blinded to the reference standard result.

For the domain “reference standard”, all studies were considered as low risk, since they correlated correctly with the target condition according to the reference standard, with definitive histology after surgical removal or follow-up with spontaneous resolution of the mass or follow-up for at least 1 year.

Regarding the domain “flow and timing”, the time elapsed between the index test and reference standard was unclear in five studies [22,23,28,29,30]. In the remaining six studies, it was considered as low risk [24,25,26,27,31,32].

Concerning applicability, all studies were considered as low risk regarding patient selection (target population: patient with an adnexal mass), index test (ultrasound), and reference standard (surgery or follow-up) domains.

### 3.4. Diagnostic Performance of O-RADS System for Classifying Adnexal Masses

Overall, the pooled estimated sensitivity, specificity, positive likelihood ratio, negative likelihood ratio, and DOR of the O-RADS system for classifying adnexal masses were 97% (95% confidence interval (CI) = 94%–98%), 77% (95% CI = 68%–84%), 4.2 (95% CI= 2.9–6.0), and 0.04 (95% CI = 0.03–0.07), and 96 (95% CI = 50–185), respectively.

Observed heterogeneity was moderate for sensitivity (I^2^ = 55.1%; Cochran Q = 22.2; *p* < 0.001) and high for specificity (I^2^ = 95.3%; Cochran Q= 214.2; *p* < 0.001). The forest plot is shown in Figure 3. Meta-regression showed that neither sample size nor malignancy prevalence explained the heterogeneity observed.

The sROC curve for diagnostic performance of the O-RADS system in classifying adnexal masses is shown in Figure 4. The area under the curve was 0.97 (95% CI: 0.96–0.98).

Fagan’s nomogram showed that O-RADS 4–5 increased the pre-test probability of ovarian malignancy from 32.0% to 66.0%, while O-RADS 1–3 decreased the pre-test probability from 32.0% to 2.0% (Figure 5).

We did not observe publication bias (*p* = 0.20).

## 4. Discussion

### 4.1. Summary of Evidence

In the present study, we performed a systematic review and meta-analysis of the O-RADS classification system using transvaginal ultrasound. We found **11** studies with available data comprising more than 4500 patients. The mean prevalence of malignancy was 32%, and benign malignant tumors more frequent in postmenopausal than in premenopausal women.

We observed that the pooled sensitivity and specificity of O-RADS system were 97% and **77%**, respectively. The vast majority of the studies reported so far were retrospective, with only one prospective study with a small sample size (*n* = 50). Additionally, only one study used the O-RADS classification based on the results of the ADNEX model, whereas most used the interpretation of ultrasound features of the adnexal masses according to IOTA criteria.

### 4.2. Limitations and Strengths

The main strength of our study is that this meta-analysis is the first to address this issue. We do believe that the methodology used is correct.

As a limitation of our meta-analysis, we consider that the number of studies included was low. Therefore, we must be cautious when interpreting the results reported herein. On the other hand, most studies used the IOTA classification based on sonographic features of adnexal masses and not the ADNEX model. Therefore, we could not compare whether both approaches offer similar diagnostic performance.

### 4.3. Interpretation of Results

Our data indicate that the O-RADS classification system offers a very high sensitivity and moderate specificity for classifying adnexal masses. However, we observed a significant heterogeneity among studies for both sensitivity and specificity. This also implies that our results should be interpreted with caution.

Regarding the quality of the studies, we found room for improvement in study design. Clearly, there is a need for more prospective studies with large series of patients.

Notwithstanding, our data can be valuable from the clinical point of view. We observed that the specificity of O-RADS classification is moderate (pooled false positive rate of 23%). This finding deserves attention, since, in most studies analyzed in this meta-analysis, the examiners involved were expert examiners. According to current evidence, a higher specificity should be expected for expert examiners when using subjective impression [5,33]. Expert examiners need to use clearly defined criteria for describing adnexal masses. However, most of the times, an expert examiner establishes their diagnostic judgment on the basis of a subjective assessment, which in turn is based on so-called “pattern recognition”. In fact, this is how the meta-analysis by Meys analyzed this issue. This meta-analysis showed that the pooled sensitivity and specificity for “subjective assessment” were 93% and 89%, respectively. On the other hand, O-RADS does not allow providing a judgment of risk malignancy on the basis of a “subjective assessment”, but instead using either the ADNEX model or IOTA lexicon. This is why we do think that O-RADS could render a poorer specificity if used by an expert examiner. However, the pooled sensitivity of O-RADS was high (97%). This is important, since few ovarian malignancies would be missed using this classification and, therefore, in nonexpert hands, this is quite relevant (in spite of a significant false positive rate). For this reason, we do think that this system would be more suitable for nonexpert examiners.

Our data must be also interpreted in the context of a comparison with other ultrasound-based and biomarker-based models. The data we obtained for O-RADS classification provided figures, in terms of sensitivity and specificity, similar to those reported for IOTA Simple Rules [33,34,35], IOTA LR2 [33,34], and the IOTA ADNEX model [36,37]. However, O-RADS classification offered better sensitivity and specificity than the risk of malignancy index (RMI) [11,33,34] and the risk of ovarian malignancy algorithm (ROMA) [11,38].

In addition, a recent consensus paper developed by several societies such as the European Society of Gynecologic Oncology (ESGO), the European Society of Gynecologic Endoscopy (ESGE), the International Society of Ultrasound in Obstetrics and Gynecology (ISUOG), and the IOTA group stated that, in spite of the O-RADS classification system not having been validated, it should be used to classify adnexal masses, and the consensus paper provided management guidelines accordingly [39]. Our results now support this recommendation.

### 4.4. Future Research Agenda

As stated above, there is a need for prospective studies in large series of patients to definitively validate the O-RADS ultrasound classification system.

Although some studies have addressed the issue of reproducibility, there is a need for more studies assessing this important issue, as well as studies assessing how the O-RADS system works in the hands of nonexpert examiners.

Lastly, the O-RADS system was developed for providing guidance in patient management. To date, no study has addressed this issue properly.

## 5. Conclusions

In conclusion, our data show that the O-RADS ultrasound system has good diagnostic performance in classifying adnexal masses. However, more and better-designed studies are needed to determine whether this system should be used as standard in the ultrasound evaluation of adnexal masses.

## Figures and Tables

**Figure 1 cancers-14-03151-f001:**
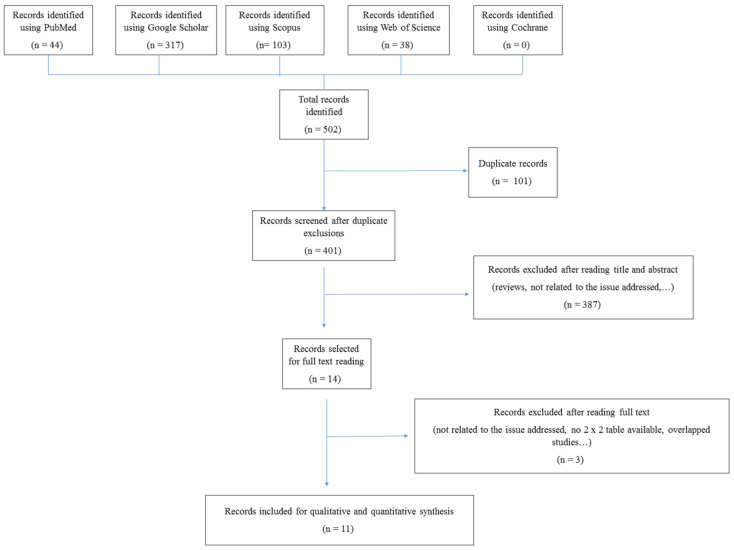
Flowchart showing study selection process, indicating the titles found in each database, as well as the exclusion process and the final number of studies ultimately included in the meta-analysis.

**Figure 2 cancers-14-03151-f002:**
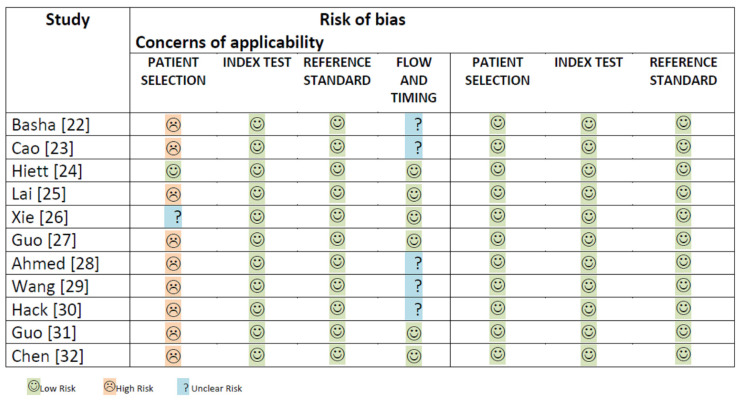
This figure shows the quality assessment (risk of bias and concerns about applicability) for all studies included in the meta-analysis.

**Figure 3 cancers-14-03151-f003:**
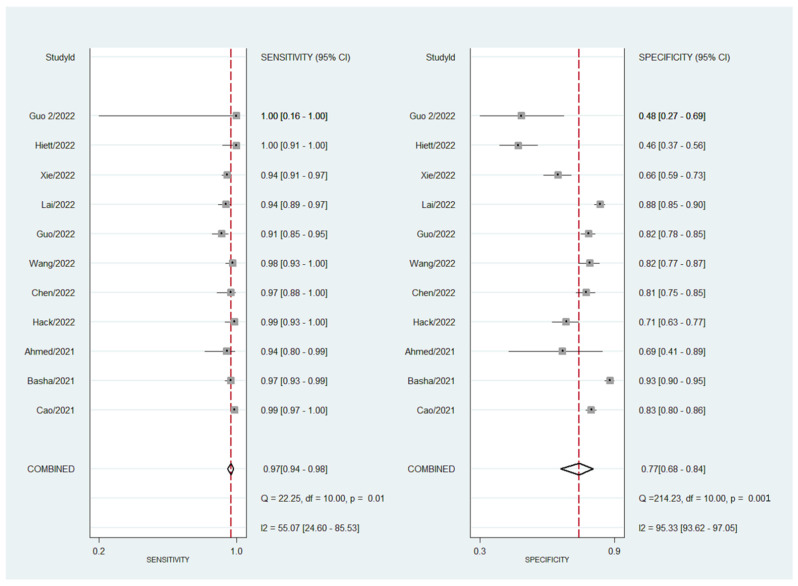
Forest plot for sensitivity and specificity for all studies using the O-RADS reporting system. Pooled sensitivity and specificity are also shown, as well as the heterogeneity found.

**Figure 4 cancers-14-03151-f004:**
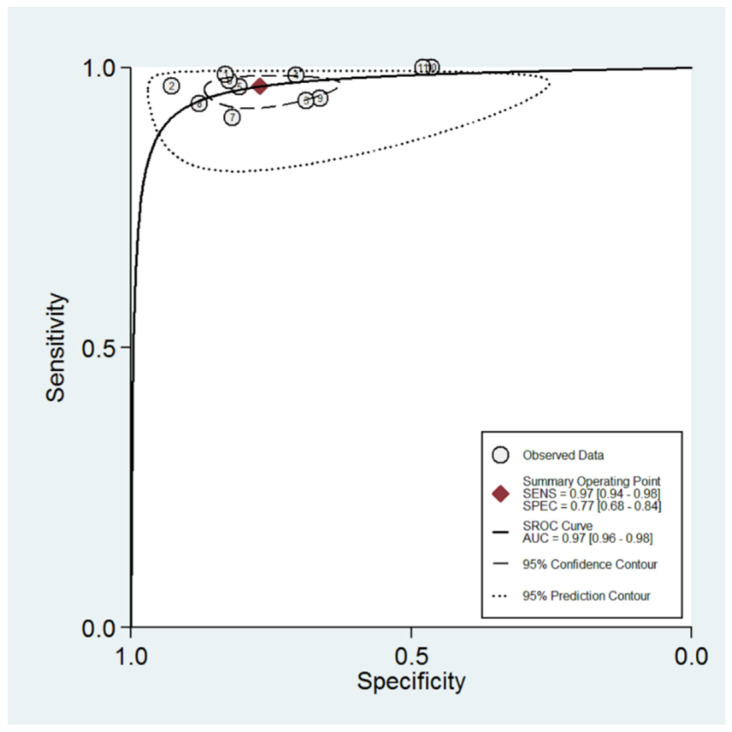
Summary ROC curve for O-RADS reporting system showing the sensitivity and specificity for each study and pooled estimation. The dashed line around the summary point estimate (red diamond) represents the 95% confidence region. The dotted line showing the 95% prediction contour corresponds to the predicted performance taking into account all individual studies.

**Figure 5 cancers-14-03151-f005:**
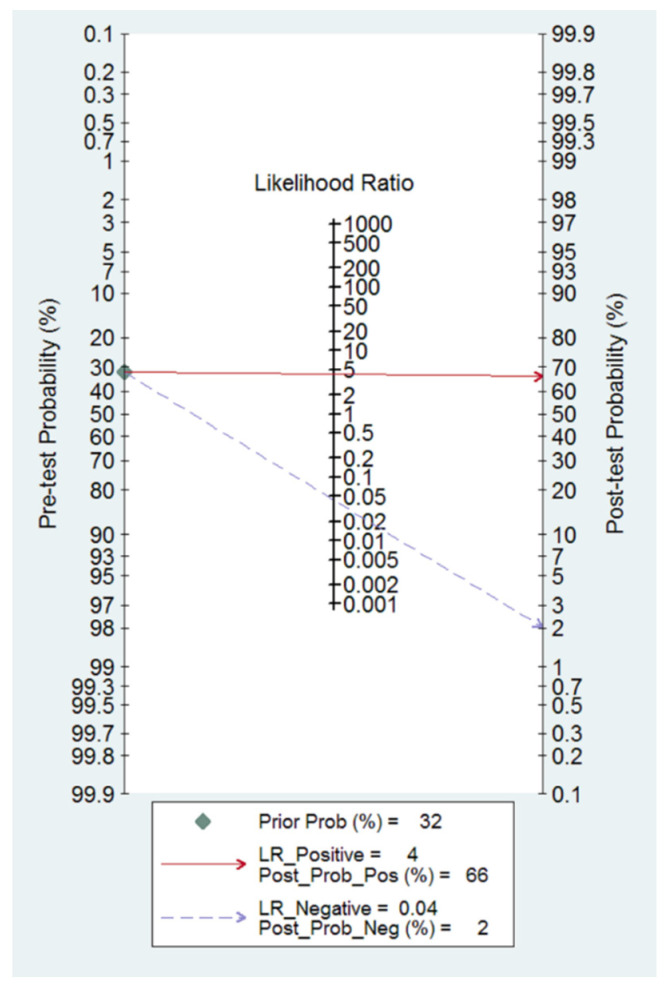
Fagan nomogram for O-RADS reporting system. It can be observed how the test changes the pre-test probability depending on a positive or negative result.

**Table 1 cancers-14-03151-t001:** Main characteristics of the studies included in the meta-analysis.

Author	Year	Country	Number of Patients	Number of Masses	Malignant Cases	Patients’ Mean Age (Years) Benign Tumors	Patients’ Mean Age (Years) Malignant Tumors	Number of Examiners	Study’s Design	Index Test	Reference Test	Time Elapsed from US to Surgery	Examiner Blinded
Basha [22]	2021	Egypt	609	647	178	NA	NA	Five	Retrospective	O-RADS IOTA features	Histology/Follow-up	NA	Yes
Cao [23]	2021	China	1035	1054	304	37	53	Two	Retrospective	O-RADS IOTA features	Histology	NA	Yes
Ahmed [28]	2021	Egypt	50	50	35	NA	NA	One	Prospective	O-RADS IOTA features	Histology/Follow-up	NA	Yes
Hiett [24]	2022	USA	150	150	40	47	48	Two	Retrospective	O-RADS ADNEX model	Histology	<180 days	Yes
Lai [25]	2022	China	734	734	170	35	48	Two	Retrospective	O-RADS IOTA features	Histology	<120 days	Yes
Xie [26]	2022	China	453	453	269	45	51	Two	Retrospective	O-RADS IOTA features	Histology	<120 days	Yes
Guo [31]	2022	China	25	25	2	NA	NA	Two	Retrospective	O-RADS IOTA features	Histology	<90 days	Yes
Guo [27]	2022	China	575	592	145	37	46	Two	Retrospective	O-RADS IOTA features	Histology/Follow-up	<30 days	Yes
Wang [29]	2022	China	345	345	128	40	52	Two	NA	O-RADS IOTA features	Histology	NA	NA
Hack [30]	2022	USA	227	262	75	NA	NA	Two	Retrospective	O-RADS IOTA features	Histology/Follow-up	NA	Yes
Chen [32]	2022	Taiwan	322	322	58	NA	NA	Two	Retrospective	O-RADS IOTA features	Histology	<120 days	Yes

NA: Information not available.

**Table 2 cancers-14-03151-t002:** Distribution of O-RADS cases according to reference standard in studies reporting this information.

Author	Year	Number of Masses	Malignant Cases	O-RADS 1–2		O-RADS 3		O-RADS 4		O-RADS 5	
				Benign	Malignant	Benign	Malignant	Benign	Malignant	Benign	Malignant
Basha [22]	2021	647	178	261	1	174	5	26	11	8	161
Cao [23]	2021	1054	304	445	1	179	3	97	51	29	249
Ahmed [28]	2021	50	35	0	0	11	2	3	15	2	19
Hiett [24]	2022	150	40	17	0	34	0	52	14	7	26
Lai [25]	2022	734	170	364	5	131	6	65	90	4	69
Xie [26]	2022	453	269	74	4	48	11	52	82	10	172
Guo [31]	2022	25	2	7	0	4	0	13	1	1	1
Guo [27]	2022	592	155	294	6	72	7	73	80	8	52
Wang [29]	2022	345	128	93	1	86	2	27	44	11	81
Hack [30]	2022	262	75	100	0	31	1	41	22	15	52
Chen [32]	2022	322	58	149	0	66	2	69	26	38	30

**Table 3 cancers-14-03151-t003:** O-RADS distribution according to histology *.

Histology	O-RADS 2	O-RADS 3	O-RADS 4	O-RADS 5	Total
Benign					
Functional cyst	33	7	6	1	47
Hemorrhagic cyst	5	4	3	0	12
Endometrioma	312	80	24	2	418
Dermoid cyst	320	98	39	4	461
Serous/mucinous cystadenoma	114	115	87	6	322
Para-ovarian cyst	7	2	1	0	10
Hydrosalpinx/TOA	20	15	17	11	63
Fibroma/fibrothecoma/thecoma	4	14	23	6	47
Struma ovarii	0	2	3	0	5
Other benign lesions	14	7	9	8	38
Malignant					
Borderline tumor	7	8	63	35	113
Epithelial carcinoma	0	0	69	245	314
Germ cell tumor	0	0	4	4	8
Sex-cord tumor	0	0	6	7	13
Metastatic tumor	0	0	10	27	37
Other malignant lesions	0	0	5	27	32
Total	839	357	383	385	1964

TOA: Tubo-ovarian abscess. * Data obtained from references [23,24,25,31].

## Data Availability

Data are available upon request.

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
