# Peer review of "Ovarian Adnexal Reporting Data System (O-RADS) for Classifying Adnexal Masses: A Systematic Review and Meta-Analysis"

_cancers, 2022, doi:10.3390/cancers14133151_

Round 1

Reviewer 1 Report

51: Transvaginal ultrasound (TVS): it is more appropriate to say Transvaginal Sonography (TVS), the abbreviation belongs to this.

English language and grammar mistakes: 57 has been proved to be on; 61 WIth the aim of to. 183-184 sentence not normal. So English should be checked!

Methods: The athours did not use EMBASE in their search strategy. Google Scholar is used. This questions the accuracy of the search. 76-78: is available upon reasonable request. The search should be described in more detail, not only upon "reasonable request".

103: overlapping dates: we chose the last published. It would have been better to contact the authors of that specific article instead of leaving out data in a meta-analysis. This decision makes one doubt the completness of the included studies.

Is this a meta-analysis or more a review? Are all individual patient data used or the results of the studies, such as sensitivity and specificity per study?

The athors conclude that more prospective studies are needed, and that is e very good conlusion! 

Author Response

  1. Transvaginal ultrasound (TVS): it is more appropriate to say Transvaginal Sonography (TVS), the abbreviation belongs to this.
    1. Thanks for this comment. Corrected as suggested.
  2. English language and grammar mistakes: 57 has been proved to be on; 61 WIth the aim of to. 183-184 sentence not normal. So English should be checked!
    1. Mistakes corrected. Regarding “183-184 sentence not normal”. Excuse us, but we are not sure to what sentence the reviewer refers to
  3. The authors did not use EMBASE in their search strategy. Google Scholar is used. This questions the accuracy of the search
    1. Certainly, not using EMBASE database could limit our search, but we do not think to the point of questioning the whole search, even more when we are assessing a quite recent model, actually published for the first time barely 30 months ago
  4. 76-78: is available upon reasonable request. The search should be described in more detail, not only upon "reasonable request".
    1. We agree this sentence has been modified. It was a mistake. Some data, such a as exclusion criteria are provided in the Revised Version. See page 4, lines 104-107
  5. 103: overlapping dates: we chose the last published. It would have been better to contact the authors of that specific article instead of leaving out data in a meta-analysis. This decision makes one doubt the completeness of the included studies.
    1. Thank you so much for this advice. We contacted the corresponding authors and we learnt that no overlapping existed indeed. Thus, two new studies have been added. We have performed new analysis. Final results did not change.
  6. Is this a meta-analysis or more a review? Are all individual patient data used or the results of the studies, such as sensitivity and specificity per study?
    1. This is a systematic review and meta-analysis. However, we did not perform individual data analysis. So, we do reflect pooled data from studies

Reviewer 2 Report

The submitted work summarizes the available evidence on the Ovarian-Adnexal Reporting Data System (O-RADS) by means of a systematic review and meta-analysis. The topic is of eminent clinical importance. Overall, the manuscript is of high quality and, after minor improvements (see below), deserves publication. The abstract is informative and sufficient. The introduction should include brief information about other diagnostic tools used in differentiation of the adnexal mass (especially biomarker-based models and multimodal tools), as the authors mention only that TVS is the first-line modality. The Methods section should provide information on inclusion and exclusion criteria for studies and methods used for data extraction and quality assessment. Please provide the Figures (particularly Fig. 4 and 5) in high resolution. The discussion is the weakest part of the manuscript: The results should be interpreted in the context of other ultrasound-based scoring systems as well as biomarker-based models.  Finally, the manuscript should be formatted according to the journal's requirements.

Author Response

Dear reviewer, thanks for your comments. We have done the following amendments:

  1. The introduction should include brief information about other diagnostic tools used in differentiation of the adnexal mass (especially biomarker-based models and multimodal tools), as the authors mention only that TVS is the first-line modality.
    1. Thanks for this comment. We have mentioned these diagnostic tools in the Revised Version. We have added a new reference (#11)
  2. The Methods section should provide information on inclusion and exclusion criteria for studies and methods used for data extraction and quality assessment.
    1. This is already explained in pages 6 to 8 on the manuscript
  3. Please provide the Figures (particularly Fig. 4 and 5) in high resolution
    1. New figures with higher resolution submitted
  4. The discussion is the weakest part of the manuscript: The results should be interpreted in the context of other ultrasound-based scoring systems as well as biomarker-based models.
    1. We have added a paragraph addressing this question (see page 18, lines 294-300)

Reviewer 3 Report

In this article, the authors performed a systematic review and meta-analysis of the diagnostic performance of the American College of Radiology’s reporting system called Ovarian-Adnexal Report Data System (O-RADS) that was introduced in 2020 in discriminating malignant (O-RADS 4-5) from benign (O-RADS 1-3) adnexal masses. Based on 9 studies comprising 7350 adnexal masses, they concluded that O-RADS offers a good sensitivity and moderate specificity for classifying adnexal masses.

Although this study is clinically significant and appears to have been conducted correctly, the clarity of the presentation needs to be improved throughout the manuscript. Specifically, the authors need to improve the flow of ideas by explaining and discussing in more detail the analyses performed and the transitions from one analysis to the next.

Line 26: indicate that the O-RADS system was introduced in 2020.  Also, mention key selection and exclusion criteria.

Line 41: the keywords should reflect more the paper’s content: adnexal masses, meta-analysis, ovarian neoplasms, O-RADS, transvaginal ultrasound

Line 56-57: incomplete sentence. Also, define all abbreviations and acronyms (e.g., ADNEX, IOTA…) at first mention in the abstract and main text.

Line 76-78: are the inclusion and exclusion criteria used those described at Line 91-101?  Please clarify. For no reasons should the readers have to request such information from the authors!!

Line 79: from which university or hospital was the Institutional Review Board?

Line 91-101: were all 3 criteria required to qualify as an eligible study?

Line 107: provide reference(s).

Line 117: what constitutes “low, high or unclear risk”? Be specific.

Line 144, 222, 239: correct “GIRADS” with “O-RADS”.

Line 147: add “Likelihood ratios”.

Line 185, 212: what pathologic type of malignancies were diagnosed?  Were they primary, recurrent or metastatic?

Line 203: what were “inappropriate exclusions”?

Line 211: change “considered” with “correlated” as in radiopathological correlation.

Line 214-216: what was the time elapsed in the different studies?

Line 235: in Figure 4, what is the difference between 95% confidence contour and 95% prediction contour?

Line 245: in Figure 6, what is “1/root(ESS)”?  Is it square root?

Line 268-269: what constitute “very high sensitivity” and “moderate specificity”?

Line 273: which improvements in study design are needed?

Line 281-283: even experts need to use clearly defined criteria, not “subjective impression”. Explain your conclusion that “O-RADS system in expert hands… for non-expert examiners”.

Line 287-288, 298-299: how is it that the O-RADS system has not been validated? Please clarify.

Table 1: indicate the age range of the patients and the number of examiners, not just multiple.

Author Response

  1. Line 26: indicate that the O-RADS system was introduced in 2020. Also, mention key selection and exclusion criteria
    1. Done as suggested
  2. Line 41: the keywords should reflect more the paper’s content: adnexal masses, meta-analysis, ovarian neoplasms, O-RADS, transvaginal ultrasound
    1. Done as suggested
  3. Line 56-57: incomplete sentence. Also, define all abbreviations and acronyms (e.g., ADNEX, IOTA…) at first mention in the abstract and main text
    1. Done as suggested
  4. Line 76-78: are the inclusion and exclusion criteria used those described at Line 91-101? Please clarify. For no reasons should the readers have to request such information from the authors!!
    1. Yes, they are the same. No change made
  5. Line 79: from which university or hospital was the Institutional Review Board?
    1. From Clinica Universidad de Navarra
  6. Line 91-101: were all 3 criteria required to qualify as an eligible study?
    1. Indicated in the text in the revised version
  7. Line 107: provide reference(s).
    1. We guess the reviewer refers to this sentence “the study period of each study was examined; if dates overlapped, we chose the latest study published”. In fact, there is no reference to provide. We describe an “a priori” situation as part of the methodology used to explain how we handled this situation.
  8. Line 117: what constitutes “low, high or unclear risk”? Be specific.
    1. We already explain this in the subsequent paragraphs. Please see the text
  9. Line 144, 222, 239: correct “GIRADS” with “O-RADS”.
    1. Corrected
  10. Line 147: add “Likelihood ratios”.
    1. Added as suggested
  11. Line 185, 212: what pathologic type of malignancies were diagnosed? Were they primary, recurrent or metastatic?
    1. Data added
  12. Line 203: what were “inappropriate exclusions”?
    1. Explained in the text.
  13. Line 211: change “considered” with “correlated” as in radiopathological correlation.
    1. Modified as suggested
  14. Line 214-216: what was the time elapsed in the different studies?
    1. Information added in table 1
  15. Line 235: in Figure 4, what is the difference between 95% confidence contour and 95% prediction contour?
    1. A
  16. Line 245: in Figure 6, what is “1/root(ESS)”? Is it square root?
    1. Yes, it is square root
  17. Line 268-269: what constitute “very high sensitivity” and “moderate specificity”?
    1. This is a subjective appreciation.
  18. Line 273: which improvements in study design are needed?
    1. Prospective studies, as stated in the subsequent sentence. No change made
  19. Line 281-283: even experts need to use clearly defined criteria, not “subjective impression”. Explain your conclusion that “O-RADS system in expert hands… for non-expert examiners”.
    1. Certainly, we agree that expert examiners need to use clearly defined criteria for describing adnexal masses. However, most of the times, an expert examiner establishes her/his diagnostic judgment based on a subjective assessment (we could say “impression”), which in turn is based on the so-called “pattern recognition”. In fact, this is how Meys’ meta-analysis analyzed this issue. This meta-analysis showed that the pooled specificity for “subjective assessment” was above 90%. On the other hand, O-RADS does not allow providing a judgment of risk malignancy based on this “subjective assessment” but using either the ADNEX model or IOTA lexicon. As we have found that pooled specificity for O-RADS is 78%, this is why we state in our manuscript that O-RADS could render a poorer specificity if used by an expert examiner. However, pooled sensitivity of O-RADS is high (97%). This is important, since few ovarian malignancies would be missed using this classification and, therefore, in non-expert hands this is quite relevant (in spite of a significant false positive rate). No change made in the manuscript
  20. Line 287-288, 298-299: how is it that the O-RADS system has not been validated? Please clarify
    1. This is the actual situation. O.-RADS has not been externally validated in well-designed prospective large series studies. Actually, this is recognized in the ESGO-ISOUG-IOTA-ESGE consensus (See this sentence from this article “However, to date, neither the triage system nor the O-RADS descriptors have been externally validated”). Non change made in the manuscript
  21. Table 1: indicate the age range of the patients and the number of examiners, not just multiple.
    1. Only one study reported patients¡ age range. Therefore, we have not added this information. The number of examiners is provided in the revised version

Round 2

Reviewer 3 Report

We appreciate your reply to the first review report.  Please address the following additional comments.

Line 15 and 269: indicate that this analysis is based on transvaginal sonography data to differentiate it from the MRI O-RADS classification.

Line 16: write in full “diagnostic odds ratio”.

Line 60-61: the sentence “For this reason… models” is incomplete.

Line 68-71: summarize briefly the five risk groups and define IOTA.

Line 70-71: which are the “several studies”? Provide references.

Line 86-87: five electronic databases are listed. Do you mean that three of the authors screened three databases each?

Fig. 1-6: provide brief descriptive legends in addition to the figure titles.

Fig. 2: the color code boxes are barely legible.  Which study had the highest risk of patient selection bias?

Line 213: do you mean that excluding cases because of poor image quality or unavailable data introduces an “inappropriate” selection bias?

Line 231: regarding the diagnostic performance of the O-RADS classification, how well did the O-RADS results correlate with the histology and/or follow-up data in Table 1? Please elaborate. 

Fig. 4: what is the difference between 95% confidence contour and 95% prediction contour?

Fig. 6: the circled reference numbers are barely visible. Please number the corresponding references in Table 1 and 2.

Line 298-302: please clarify this section in the text of the manuscript as you did in your response letter. Regarding Meys’ meta-analysis (Ref. 33), was it based on transvaginal sonography data, and how well did the imaging results correlate with the histology and/or follow-up data?

Line 309: how well did the O-RADS classification based on transvaginal sonography data compare with the MRI O-RADS classification?  Please elaborate.

Line 313-314: was the O-RADS classification system (based on transvaginal sonography or MRI data) validated based on histology and/or follow-up data?

Author Response

Response to reviewer #3

  1. Line 15 and 269: indicate that this analysis is based on transvaginal sonography data to differentiate it from the MRI O-RADS classification.
    1. Added, as suggested
  2. Line 16: write in full “diagnostic odds ratio”.
    1. Modified as suggested
  3. Line 60-61: the sentence “For this reason… models” is incomplete.
    1. Sentence re-worded
  4. Line 68-71: summarize briefly the five risk groups and define IOTA
    1. Done as suggested
  5. Line 70-71: which are the “several studies”? Provide references.
    1. Done as suggested
  6. Line 86-87: five electronic databases are listed. Do you mean that three of the authors screened three databases each?
    1. One author searched Pubmed and Cochrane, a second author searched Scopus and Web of Science and a third author searched Google Scholar. Do you want us to indicate so in the text?
  7. 1-6: provide brief descriptive legends in addition to the figure titles.
    1. Done as suggested
  8. 2: the color code boxes are barely legible. Which study had the highest risk of patient selection bias?
    1. We agree with reviewer. Figure 2 has been change to better show the qualitative analysis of each study
  9. Line 213: do you mean that excluding cases because of poor image quality or unavailable data introduces an “inappropriate” selection bias?
    1. This a way for selecting the “best cases”, which clearly is a selection bias. No modification done
  10. Line 231: regarding the diagnostic performance of the O-RADS classification, how well did the O-RADS results correlate with the histology and/or follow-up data in Table 1? Please elaborate.
    1. Do you mean adding data of TP, TN, FP and FN for each study in table 1? If so, table 1 would be quite large. Please, reconsider. If, in any case, you consider this information as relevant, we would add a new table instead. Actually, performance of each study is reported in figure 3 (forest plot)
  11. 4: what is the difference between 95% confidence contour and 95% prediction contour?
    1. 95% confidence contour is the 95% confidence interval for the pooled estimation of sensitivity (97%) and specificity (77%) (red diamond in the figure). 95% prediction contour corresponds to the predicted performance taking into account all individual studies
  12. 6: the circled reference numbers are barely visible.
    1. Certainly, the reviewer is right. We have tried to change figure size for better visualization of reference numbers but we could not. So, we have decided to remove this figure. Actually, it is not relevant.
  13. Please number the corresponding references in Table 1 and 2.
    1. Done as suggested
  14. Line 298-302: please clarify this section in the text of the manuscript as you did in your response letter.
    1. Done as suggested
  15. Regarding Meys’ meta-analysis (Ref. 33), was it based on transvaginal sonography data, and how well did the imaging results correlate with the histology and/or follow-up data?
    1. Yes, Meys’ meta-analysis address studies using transvaginal ultrasound. Data reported in the text
  16. Line 309: how well did the O-RADS classification based on transvaginal sonography data compare with the MRI O-RADS classification? Please elaborate.
    1. RMI refers to a risk estimation based on menopausal status, ultrasound findings and CA-125 levels, not to MRI. Certainly, there is a O-RADS MRI classification. But to the best of our knowledge no study has compared O-RADS ultrasound and O-RADS MRI. No change made in the manuscript.
  17. Line 313-314: was the O-RADS classification system (based on transvaginal sonography or MRI data) validated based on histology and/or follow-up data?
    1. No, not in well-designed prospective studies, as we state in the manuscript. No change made in the text.

Round 3

Reviewer 3 Report

Thank you for your revisions. 

Please spell check your document.  Line 17, 34, 156…: correct “odd” with “odds”.

Line 149-152: since the five risk groups are described in the introduction (line 69-72), you could refer to the introduction for the definition of the groups. However, note that the interval percentages listed are different.

Line 194-195: since all studies reported the histology of the masses, please mention the most common histopathological diagnoses of borderline tumors (e.g., epithelial borderline neoplasm), primary ovarian cancers (e.g., epithelial carcinoma) and metastatic cancers (e.g., metastatic endometrial carcinoma) in the text.  Describe the distribution of the histopathological diagnoses among the 9 selected studies.

Line 197: although all studies reported the distribution of masses according to menopausal status, that important information was not taken into consideration in the analysis and discussion. Please include the menopausal status in your analysis.

Line 213, 276: please make clear that Fig. 2 has been replaced and that Fig. 6 has been deleted.  Describe in the text the results of Fig. 6.

Line 239: regarding the diagnostic performance of the O-RADS system, how do the different histopathological diagnoses mentioned in the 9 selected studies correlate with the sensitivity and specificity of the O-RADS system for classifying adnexal masses?  Please summarize and discuss these results in the text.

Line 249: indicate the full p value of the combined specificity, not “p=0.00”.

Line 261-263: please add in the figure legend the text of your reply letter about confidence contour and prediction contour.

Line 320: mention Ref. 33.

Author Response

Thanks for your comments. The following amendments have been made

  1. Please spell check your document. Line 17, 34, 156…: correct “odd” with “odds”
    1. Corrected
  2. Line 149-152: since the five risk groups are described in the introduction (line 69-72), you could refer to the introduction for the definition of the groups. However, note that the interval percentages listed are different.
    1. Thanks for this comment. We refer to the introduction and delete the corresponding risks estimates in this sentence
  3. Line 194-195: since all studies reported the histology of the masses, please mention the most common histopathological diagnoses of borderline tumors (e.g., epithelial borderline neoplasm), primary ovarian cancers (e.g., epithelial carcinoma) and metastatic cancers (e.g., metastatic endometrial carcinoma) in the text. Describe the distribution of the histopathological diagnoses among the 9 selected studies
    1. This information is not fully available in all studies. For example, in six studies (Cao, Guo, Hack, Hiett, Chen and Wang) do not report information about the type of borderline tumor. In all studies, information about the primary tumor in case of metastatic tumors is not provided. So, we provide a summary of epithelial primary ovarian cancer, non-epithelial ovarian cancer and benign tumors
  4. Line 197: although all studies reported the distribution of masses according to menopausal status, that important information was not taken into consideration in the analysis and discussion. Please include the menopausal status in your analysis.
    1. We guess the reviewer refers to menopausal status and outcome of the lesions according to the reference standard. We added this information in the Revised Version.
  5. Line 213, 276: please make clear that Fig. 2 has been replaced and that Fig. 6 has been deleted. Describe in the text the results of Fig. 6.
    1. It is clearly stated both issues. In lines 209 and 265, respectively
  6. Line 239: regarding the diagnostic performance of the O-RADS system, how do the different histopathological diagnoses mentioned in the 9 selected studies correlate with the sensitivity and specificity of the O-RADS system for classifying adnexal masses? Please summarize and discuss these results in the text
    1. This information was only available from four studies. We report these data as a new table (Table 3). Notwithstanding, we should careful when interpreting this table because it could be misleading. For example, taking into account these 1964 tumors, no invasive primary ovarian malignancy oi metastatic tumor was classified as O-RADS2 or O-RADS 3. Only 15 BOTs cases were classified as O-RADS 2 or 3. However, if we look at table 1 were all cases are taken into account, we can observe that a higher number of malignant tumors were actually classified as O-RADS 2 or 3.  Similarly, might happen with benign tumors. The reader could have the impression that a lower number of benign tumors can be classified as O-RADS 4 or 5 as compared to data from table 1. Please consider our comment for avoiding misleading information.
  7. Line 249: indicate the full p value of the combined specificity, not “p=0.00”.
    1. We guess the reviewer refers to the figure 3. This is how STATA delivers the figure. We have modified the figure to add “p=0.001”
  8. Line 261-263: please add in the figure legend the text of your reply letter about confidence contour and prediction contour.
    1. Done
  9. Line 320: mention Ref. 33.
    1. Already mentioned. Highlighted in the Revised Version